# The Role of Diet and Physical Activity in Shaping COVID-19 Severity: Design, Validation, and Application of a Retrospective Questionnaire

**DOI:** 10.3390/healthcare12080813

**Published:** 2024-04-10

**Authors:** Francisco Vásquez-Aguilar, Marcela de Jesús Vergara-Jiménez, Oscar G. Figueroa-Salcido, Jesús Gilberto Arámburo-Gálvez, Feliznando Isidro Cárdenas-Torres, Noé Ontiveros, Erika Martínez-López, Elisa Barrón-Cabrera

**Affiliations:** 1Facultad de Ciencias de la Nutrición y Gastronomía, Universidad Autónoma de Sinaloa, Culiacan 80010, Sinaloa, Mexico; vasquezfa95@gmail.com (F.V.-A.); mjvergara@uas.edu.mx (M.d.J.V.-J.); oscar.figueroa@uas.edu.mx (O.G.F.-S.); gilberto.aramburo@uas.edu.mx (J.G.A.-G.); feliznando@uas.edu.mx (F.I.C.-T.); 2Laboratorio de Análisis Clínicos e Investigación (LACIUS, URS), Departamento de Ciencias Químico, Biológicas y Agropecuarias (DC-QB), Facultad de Ciencias Biológicas y de la Salud, Universidad de Sonora, Navojoa 85880, Sonora, Mexico; noe.ontiveros@unison.mx; 3Instituto de Nutrigenética y Nutrigenómica Traslacional, Departamento de Biología Molecular y Genómica, Centro Universitario de Ciencias de la Salud, Universidad de Guadalajara, Guadalajara 44430, Jalisco, Mexico; erika.martinez@academicos.udg.mx

**Keywords:** COVID-19, physical activity, diet, lifestyle, questionnaire

## Abstract

After the global challenges posed by COVID-19, researchers strived to identify risk factors for severe cases, which lead to various complications—including death. Lifestyle modifications, such as implementing a healthy diet and recommended physical activity, have been shown to be protective against severe COVID-19 cases. Despite an association of a plant-based diet with reduced COVID-19 severity, specific dietary characteristics have not been identified. Also, the methodology for measuring physical activity is variable, with studies overlooking the intensity or the habit components of physical activity. To bridge this gap, our study designed, validated, and applied a retrospective questionnaire with aims of exploring the relationship between lifestyle factors, specifically diet and physical activity, and severe COVID-19. We considered the intensity and years of physical activity habit, which is a limitation of other questionnaires. Results reveal associations of age and BMI with severe COVID-19. An excessive sugar diet was found to be associated with severe COVID-19 and increased symptom duration. We also observed an inverse relationship pattern of moderate- and vigorous-intensity physical activity across case severity, which is absent in walking physical activity. This study lays a foundation for research aiming to identify lifestyle factors that prevent severe COVID-19 cases.

## 1. Introduction

The emergence of the coronavirus disease 2019 (COVID-19), driven by the SARS-CoV-2 virus, had a significant adverse impact on societies worldwide. The rapid spread of this disease caused unprecedented challenges to global public health systems. As of the current date, COVID-19 has reached about 800 million cases and caused 7 million deaths globally, indicating a prevalence rate of 9.7% and a case-fatality rate of 0.9% [1]. Severe cases of COVID-19 are associated with complications such as pneumonia, acute respiratory distress syndrome, organ failure, and mortality [2]. Identified high risk factors for severe COVID-19 include age, obesity, type 2 diabetes, chronic kidney disease, chronic obstructive pulmonary disease, and coronary atherosclerosis [3,4,5]. Consequently, understanding and managing factors contributing to the severity of COVID-19 cases would allow for the prevention of its related complications. 

Lifestyle modifications, including a healthy diet and recommended levels of physical activity (PA) [6], have proven to be effective tools for improving outcomes of chronic conditions associated with severe forms of COVID-19—such as type 2 diabetes and cardiovascular diseases [7,8]. Also, a plant-based diet has been associated with reduced COVID-19 severity [9], but further research is needed to identify specific dietary characteristics influencing the severity of COVID-19. Furthermore, PA has been related to a reduction in severe COVID-19 outcomes such as hospitalization, ICU admission, and mortality [5,10,11,12,13,14,15]. However, there is a high variability in the methodology for measuring PA [10,11,12], limiting the quality of a quantitative synthesis of results (meta-analysis). Several studies use the International Physical Activity Questionnaire (IPAQ) [11,13,15] as a PA measuring tool. The IPAQ is limited to the assessment of the PA pattern of the week preceding the questionnaire application [16], overlooking the PA habit of the individual. Other studies focus on the frequency and duration of PA, ignoring the intensity component of PA [12]. 

In this context, our study designed, validated, and applied a retrospective questionnaire with aims of exploring the relationship between lifestyle factors, specifically diet and PA, and the severity of COVID-19. We examined the association between specific dietary characteristics and COVID-19 severity. We also considered the intensity component of PA beyond its frequency and duration, and integrated items to account for the temporal limitation of the IPAQ (focus on the most recent week of PA) [16]. This article serves as a hypothesis-generating piece for more comprehensive research aiming to assess the association between lifestyle and severe COVID-19.

## 2. Materials and Methods 

### 2.1. Questionnaire Design

The questionnaire (see Appendix A) was designed to identify PA and eating patterns in subjects who developed severe COVID-19. A retrospective questionnaire was applied in a cross-sectional fashion. The questionnaire was divided into the following six sections: (1) demographic information without revealing identity (age, sex, nationality, occupation, and education); (2) COVID-19 diagnosis and disease severity (diagnostic method, symptoms, hospitalization, number of days with symptoms, oxygen support); (3) medical history (particularly chronic medical conditions); (4) Diet and social history (type of diet, food cooking habits, eating schedule, water consumption, tobacco smoking, alcohol consumption, and sleep habits); (5) PA activity (based on the IPAQ short version questionnaire plus additional items addressing the type, frequency, intensity, session duration, and years of PA habit); and (6) vaccination (type of vaccine and received doses). The questionnaire comprised 56 items, but some questions could be skipped if the previous answer was negative. The questions were kept as short and simple as possible, and technical terms were avoided.

### 2.2. Questionnaire Validation

The questionnaire was validated in the following order: content validity, face validity, and test reproducibility. A Spanish native speaker expert panel assessed the validity of the content. The panel consisted of 11 health sciences professionals involved in research, including sports scientists, physicians, and nutritionists. The panel evaluated the relevance, clarity, writing, terminology, and format of each item using a numerical scale from 1 (unacceptable) to 5 (excellent) [17]. A content validity coefficient (CVC) was employed to determine the concordance of the answers among experts [17]. An item was accepted if it achieved a CVC ≥ 0.80. Items with a score below this threshold were excluded from the questionnaire.

A cohort of healthy Spanish native speakers from the northwest of Mexico assessed the face validity. This cohort was not included in the sample. Participants evaluated the clarity and comprehension of each questionnaire item. To assess clarity, a three-point ordinal scale was utilized: 1 (incomprehensible), 2 (difficult to understand), and 3 (clear and comprehensible) [18]. Comprehension was evaluated using a numerical scale ranging from 0 (very difficult to understand) to 10 (very easy to understand) [18]. Furthermore, a comment section was included, allowing participants to suggest improvements for each item. The agreement among participants’ responses was analyzed using Kendall’s W coefficient of concordance. Items were considered clear and comprehensible if they achieved both a clarity score of ≥2 and a comprehension score of ≥7, coupled with Kendall’s W coefficient of ≥0.70. These items were not subject to further rewording. Additionally, the readability of each questionnaire item was evaluated by the Flesch-Szigriszt test [19]. The INFLESZ scale was employed to interpret the results [20]. The readability test was carried out using the INFLESZ 1.0 software (Granada, Granada, Spain). A score of ≥60 was established as the benchmark for considering the questionnaire items as readable.

The reproducibility of the questionnaire was evaluated through a test–retest analysis conducted on a cohort of individuals who had recovered from COVID-19. The participants were instructed to complete the questionnaire on two different occasions, with a minimum interval of two weeks between the first and second administrations. Cohen’s kappa coefficient test was employed to analyze the consistency of the answers between the first and second administrations of the questionnaire. The time to answer the questionnaire was also recorded. 

### 2.3. Questionnaire Application

The questionnaire was digitized, and data were collected using the Survey Monkey platform. Participants were recruited through a hyperlink disseminated on social media, targeting the general population residing in Culiacan, Mexico. Data were collected from November 2022 to January 2023. Individuals aged 18 years or older who self-identified as having a history of COVID-19 disease completed the questionnaire. Only participants who concluded the entire questionnaire were included in the study. All participants provided informed consent for their involvement in this study.

### 2.4. Participants

The questionnaire was administered to 296 subjects, of whom 197 met the inclusion criteria. These criteria included either receiving a clinical diagnosis of COVID-19 by a physician or undergoing a laboratory test that yielded a positive result for COVID-19. This information was self-reported through the questionnaire. Consequently, the final sample comprised 197 participants. COVID-19 severity was classified based on the question “How do you consider the severity of your illness?” Subjects requiring supplemental oxygen were categorized as having severe COVID-19 [21], even if they reported a different perception of their severity. 

### 2.5. Measuring Lifetime Physical Activity and Sedentary Patterns

Lifetime PA patterns were obtained from the product of the following variables: metabolic equivalent task (MET) according to PA intensity; duration (minutes); frequency (days per week); and PA habit (years of PA pattern). PA intensities were categorized as walking (3.3 METs), moderate PA (4.0 METs), and vigorous PA (8.0 METs). The sedentary lifetime pattern was obtained from the product of sedentary time (hours/day) and sedentary habit (years of sedentary pattern). PA intensities, frequencies, and duration values were derived from the PA section of our questionnaire. MET intensity values align with recommended ranges from the Physical Activity Guidelines for Americans 2nd Ed. and the Compendium of Physical Activity [6,22,23].

### 2.6. Ethical Considerations 

The design, validation, and application of the retrospective questionnaire was performed from February 2022 to January 2023. This study was approved by the ethics committee of the Faculty of Nutrition and Gastronomy of the “Universidad Autónoma de Sinaloa” (Registration number CE-FCNYG-2022-FEB-002). This study was carried out according to the Declaration of Helsinki, and does not represent any physical, emotional, or occupational risk. All the participants accepted and voluntarily answered the questionnaire; however, signed informed consent was not required. The questions were designed to avoid revealing the identity of the participants and information about the researchers in charge of the study was provided. 

### 2.7. Statistical Analysis

A Shapiro–Wilk test was used to evaluate data normality and a Levene’s test to verify the homogeneity of variables. Quantitative variables are expressed as mean ± standard deviation (SD) or standard error of the mean (SEM) when the analysis was adjusted for co-variables. Pearson coefficients were used to analyze the correlation of quantitative variables. Statistical differences between groups were analyzed using a one-way ANOVA test, and post hoc tests were performed according to the homogeneity of variance. Qualitative variables are expressed as frequencies (n) and percentages (%). Associations between qualitative variables were analyzed by the application of a chi-square test. A *p* value < 0.05 was considered statistically significant. The G*Power 3.1.9.7 software was utilized to calculate a sample size sufficient for a power of 0.85. All statistical analyses were performed using the software SPSS v.29.0 (IBM, Chicago, IL, USA) and GraphPad Prism 10 software was used to create figures.

## 3. Results

### 3.1. Questionnaire Validation 

A total of 11 health professionals assessed the relevance, clarity, writing, terminology, and format of each questionnaire item. All the evaluated parameters had scores higher than 4 (Table 1), meaning that the items are clear, relevant, and have an adequate terminology and format for the target population. We included items with a CVC ≥ 0.80. The average CVC for the whole questionnaire was 0.90 (CVCt, Table 1), indicating an adequate validity and agreement among health professionals’ scores for each parameter.

In total, 53 participants (37 females, 16 males; 18–54 years old) assessed the clarity and comprehension of each questionnaire item. The average clarity score was 2.94 ± 0.04, classifying the questionnaire as clear. A strong concordance among participants for clarity scores was obtained (Kendall’s W value: 0.929), suggesting a high agreement among participants’ responses. For comprehension assessment, a continuous numerical scale ranging from 0 (very difficult to understand) to 10 (very easy to understand) was employed. We obtained a comprehension score of 9.81 ± 0.20 with a strong agreement among participants’ scores (Kendall’s W value: 0.716). No further rewording was suggested for any items evaluated. The Flesch-Szigriszt readability score was 64.88, which is considered as normal readability. 

The consistency between the responses from the first and second questionnaire application was assessed by Cohen’s k coefficient. A total of 25 participants (18 females, 7 males; 18–69 years old) answered the questionnaire twice at different times. Cohen’s k coefficient was 0.755, which is interpreted as substantial agreement. Overall, the questionnaire was classified as clear and comprehensible by participants and generates consistent responses across time. 

### 3.2. Description of the Study Sample

#### Social and Demographic Characteristics 

Our study sample (n = 197) was composed of Mexican mestizos with a mean age of 30.30 ± 12.05 years and BMI of 25.50 ± 4.21 kg/m^2^ (Table 2). Sex was distributed as 66.0% females and 34.0% males, most of them being single (62.4%). Education wise, 38.1% were students, 32.5% were employees, and the majority (69.5%) were taking courses or had finished a bachelor’s degree. The rest of the sociodemographic characteristics of the sample are illustrated in Table 2.

Regarding the COVID-19 profile (Table 3), 85.30% of the subjects were diagnosed using a laboratory test (PCR, antigen, antibody), and the rest were diagnosed solely by clinical evaluation (14.70%). Most of the individuals were diagnosed from 2020 to 2022 (98%). COVID-19 severity was classified as mild (n = 91, 46.2%), moderate (n = 89, 45.2%), and severe (n = 17, 8.6%). The participants had COVID-19 symptoms for a mean of 8.57 ± 5.72 days and lost 1.48 ± 3.7 Kg during their COVID-19 infection. The rest of the COVID-19 profile including the distribution of symptoms, comorbidities, and vaccinations are illustrated in Table 3.

### 3.3. Significant Correlations among Study Participants 

#### Social and Demographic Characteristics 

We found Significant correlations among the following quantitative variables: Age, BMI, days with symptoms, and post-COVID-19 weight loss. Table 4 illustrates their Pearson correlation coefficient (r), and their level of significance.

### 3.4. Differences per COVID-19 Severity

We conducted one-way ANOVAs to assess differences in COVID severity levels according to the following variables: age, BMI, days with symptoms, and post-COVID-19 weight loss (Figure 1). ANOVAs revealed a statistically significant difference in the subject’s age (*p* = 0.020), BMI (*p* = 0.048), days with symptoms (*p* < 0.001), and post-COVID-19 weight loss (*p* < 0.001) across COVID-19 severity levels. Post hoc tests were conducted to assess between group differences (Figure 1).

#### 3.4.1. Dietary Patterns

An excessive sugar diet was found to be associated with severe COVID-19. Figure 2 illustrates how individuals with severe COVID-19 (35%) were more likely to report adherence to an excessive sugar diet when compared with moderate (15%) and mild (11%) COVID-19 cases. A statistically significant difference across COVID-19 severity levels (*p* = 0.034) was found. 

We also found an association between an excessive sugar diet and an increased symptom duration. Figure 3 illustrates adjusted means for days with COVID-19 symptoms according to the presence or absence of an excessive sugar diet. We adjusted for the following variables: age, BMI, and post-COVID-19 weight loss. A significant difference (*p* = 0.016) was found between the presence of an excessive sugar diet ($\bar{x}$ = 10.9 ± 1.0 days) and an absence of an excessive sugar diet ($\bar{x}$ = 8.12 ± 0.4 days).

#### 3.4.2. Physical Activity Patterns

An ANOVA test was conducted for each independent variable in Figure 4, revealing overall non-significant differences (*p* > 0.05). We observed a potential inverse relationship pattern between total lifetime PA and COVID-19 severity levels (Figure 4A; mild (mean: 20,448, SE: 9834); moderate (mean: 10,731, SE: 3636); severe (mean: 2593, SE: 1105). Walking lifetime PA demonstrated an unclear relationship with COVID-19 severity [Figure 4B: mild (mean: 2750, SE: 1205); moderate (mean:1093, SE: 367); severe (mean: 1528, SE: 947)]. Moderate lifetime PA [Figure 4C; Mild (mean: 2890, SE: 1328); moderate (mean: 1407, SE: 573); severe (mean: 299, SE: 164)] and vigorous lifetime PA [Figure 4D; mild (mean: 6336, SE: 3437); moderate (mean: 3220, SE: 1311); severe (mean: 252, SE: 258)] seem to have an inverse relationship pattern with COVID-19 severity levels. Finally, the sedentary lifetime pattern might have a proportional relationship with COVID-19 severity [Figure 4E; mild (mean: 19.6, SE: 5.1); moderate (mean: 21.6, SE: 7.9); severe (mean: 23.7, SE: 11.3)].

## 4. Discussion

This study presents the strong validation scores of the questionnaire we designed and then applied (Table 1). We found associations between severe COVID-19 and age, BMI, days with symptoms, and post-COVID-19 weight loss (Figure 1). Individuals with a high-sugar diet were linked to severe COVID-19 (Figure 2) and reported an increased symptom duration (Figure 3). We also observed an inverse relationship pattern between COVID-19 severity levels and total, moderate, and vigorous lifetime PA, which is absent in walking lifetime PA (Figure 4).

Lifestyle modifications, particularly maintaining a healthy diet and engaging in PA, have been shown to improve outcomes related to chronic conditions associated with severe forms of COVID-19, such as type 2 diabetes and cardiovascular diseases [7,8]. Despite evidence suggesting an association of a plant-based diet with milder COVID-19 outcomes [9], we have limited understanding of the specific dietary characteristics influencing COVID-19 severity. Moreover, PA has been associated with a decrease in severe COVID-19 outcomes such as hospitalization, ICU admission, and mortality [10,11,12], even when controlling for other known risk factors such as age and chronic diseases [4,5,13,14,15]. The lack of recommended PA has been suggested as the most important modifiable risk factor for severe COVID-19 [5]. However, the methodology for measuring PA is heterogeneous, [10,11,12], limiting the synthesis of results (meta-analysis). The IPAQ is widely used for measuring PA [11,13,15], but is limited to the assessment of the PA pattern of the week preceding the questionnaire application [16]—overlooking the PA habit of the individual. Also, some studies focus on the frequency and duration of PA, ignoring the pivotal intensity component of PA [12], as emphasized in the most recent PA guideline for Americans [6]. 

In this context, we designed, validated, and applied a questionnaire to explore the relationship between lifestyle factors, specifically diet and PA, and the severity of COVID-19 cases. We aimed to identify specific dietary characteristics associated with COVID-19 severity. Regarding PA, our questionnaire considers the frequency, duration, and intensity of PA. We also acknowledged the temporal limitation of the IPAQ [16], which assesses the most recent week of PA assessment, by integrating an item that explores the years of PA habit. 

Our study illustrates the interplay between age, BMI, days with symptoms, and post-COVID-19 weight loss, revealing positive correlations among these factors (Table 4). We also found an association between severe COVID-19 and age; BMI; days with symptoms; and post-COVID-19 weight loss (Figure 1), confirming previous findings of age and obesity as risk factors for severe COVID-19 [4,5].

In line with the previous evidence associated with a healthy diet, such as a plant-based diet with milder COVID-19 outcomes [9], we identified an association between an excessive sugar diet and COVID-19 severity (Figure 2) and symptom duration (Figure 3). The link between a high-sugar diet and severe COVID-19 has been hypothesized in the literature [24] and is supported by our results (Figure 2 and Figure 3). A high-sugar diet has been related to elevated inflammatory markers [25], and chronic systemic inflammation has been linked to the development of chronic diseases associated with severe COVID-19 [3,4,5,26]. Therefore, this may explain the association of a high-sugar diet with severe COVID-19. 

Concerning PA, our results seem to demonstrate an inverse relationship pattern between severe COVID-19 and total, moderate, and vigorous lifetime PA (Figure 4A,C,D). Notably, no inverse relationship pattern is observed in walking lifetime PA (Figure 4B). These difference in patterns may explain the potential additive benefits of moderate to high-intensity PA on COVID-19 severity when compared to the benefits of walking or low intensity PA. The intensity of the PA seems to be a factor to consider in assessing PA protective effects against severe COVID-19, as it has been suggested for general health benefits [6]. Additionally, a slight proportional relationship pattern was observed between sedentary lifetime PA and COVID-19 severity levels (Figure 4E), in line with evidence that increased sedentary behavior increases all-cause mortality [6]. Although the relationships of lifetime PA and COVID-19 severity are not statistically significant (Figure 4), they highlight the need for further research into the role of both intensity and PA habits on the severity of COVID-19. The need for specific recommendations on the duration, frequency, habit, and intensity of PA for protection against infectious diseases has been expressed before [27]. There is no consensus on whether high-intensity PA is better than moderate-intensity PA for protecting against severe COVID-19. Some authors have observed a linearity in PA doses and reduced COVID-19 severity [4], but others have shown a flattening in the dose–response curve [12], even when considering cardiorespiratory fitness [14]. With recent studies concluding that vigorous PA does not increase the risk of infections [28], as it was once proposed [29], more research is needed to assess the differences between moderate- and vigorous-intensity PA regarding their protection against severe COVID-19.

Despite these valuable insights, it is crucial to acknowledge our study limitations. A self-reported retrospective questionnaire introduces misreporting and recall biases. The elapsed time from COVID-19 diagnosis to the application of the questionnaire had a mean of 17.7 ± 8.7 months. Therefore, to reduce recall bias, we suggest integrating a limited amount of elapsed time from COVID-19 diagnosis as an inclusion criterion of future studies. Moreover, further research on short-term vs. long-term recall would increase the validity of the retrospective questionnaire by elucidating on the adequate elapsed time for questionnaire application. Additionally, the cross-sectional design of our study and a sample size limited to a pilot study constrain causal inferences. Nonetheless, our study contributes valuable insights into the relationship between lifestyle factors and the severity of COVID-19. 

## 5. Conclusions

In conclusion, our study introduces a validated questionnaire overcoming the PA assessment of temporal limitations; it reveals associations of severe COVID-19 with age, BMI, and a high-sugar diet; and it hints for a potential additional protection against severe COVID-19 from performing moderate- to vigorous-intensity PA compared with walking PA. While preliminary, our findings lay the groundwork for further research to inform public health strategies, aiming to prevent severe cases of future COVID-19 outbreaks and enhance overall population health. 

## Figures and Tables

**Figure 1 healthcare-12-00813-f001:**
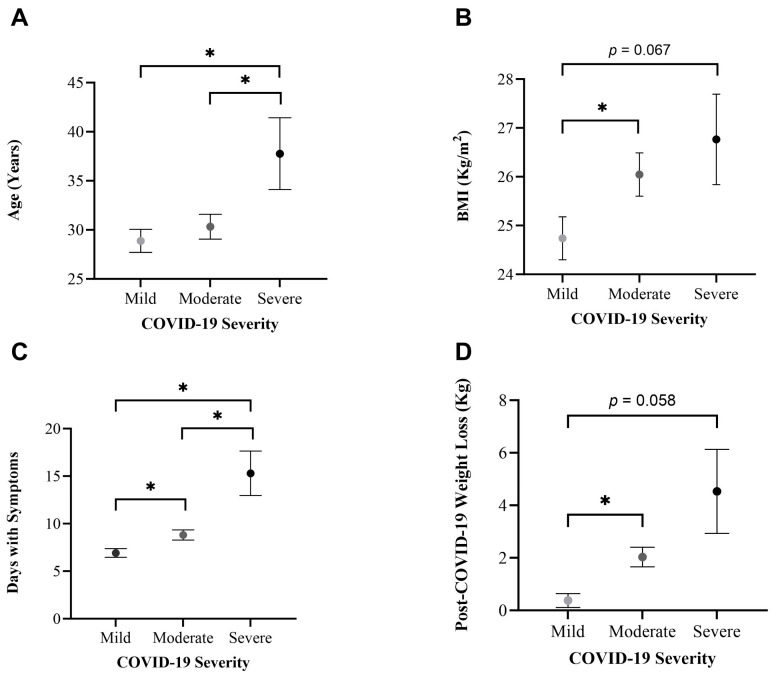
Line graphs illustrating mean and standard error of (**A**) age, (**B**) BMI, (**C**) days with symptoms, and (**D**) post-COVID-19 weight loss across COVID-19 severity levels. ANOVAs were conducted and yielded overall significant differences across COVID-19 severity levels. Post hoc * *p* < 0.05. Exact Pot Hoc *p* values demonstrating trends are included.

**Figure 2 healthcare-12-00813-f002:**
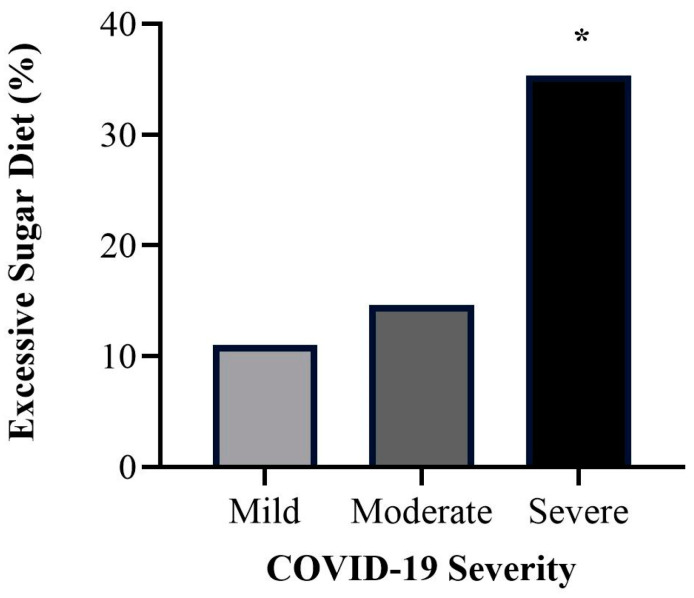
Bar graph demonstrating the percentage of subjects referring to an excessive sugar diet across COVID-19 severity levels. A chi-squared test was performed (*p* = 0.034). * = highest difference between observed and expected frequencies.

**Figure 3 healthcare-12-00813-f003:**
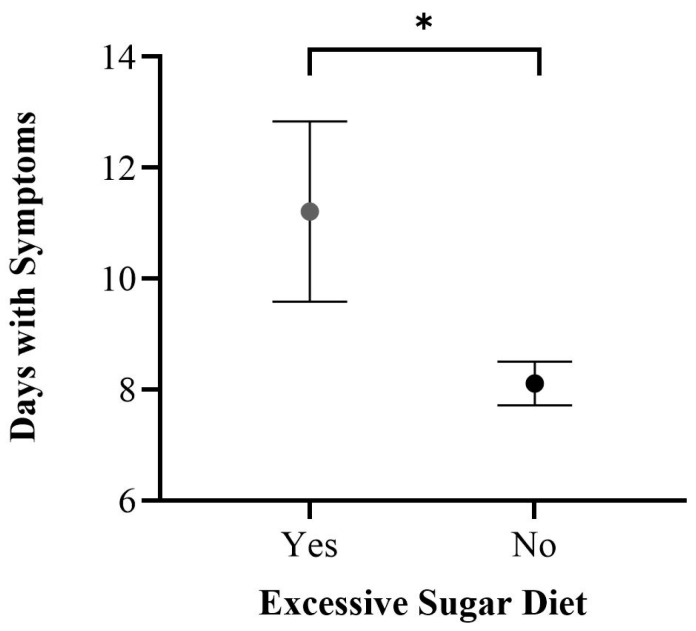
Adjusted means for days with COVID-19 symptoms according to an excessive sugar diet. A significant difference (*) was found between adjusted means (*p* = 0.016). Adjusted by age, BMI, and post-COVID-19 weight loss.

**Figure 4 healthcare-12-00813-f004:**
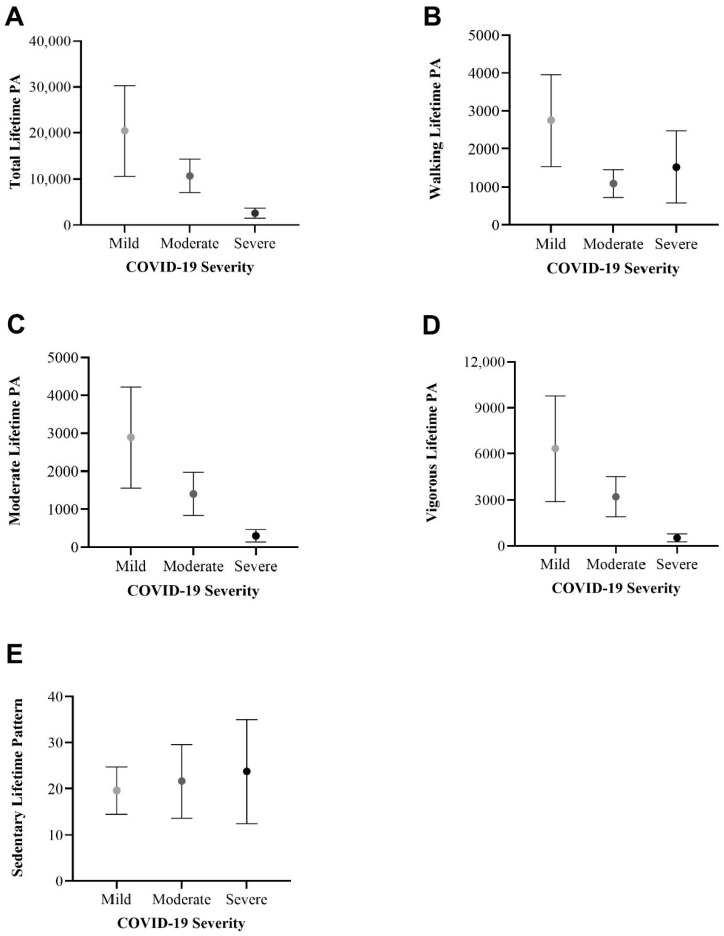
Line graphs of (**A**) total lifetime PA, (**B**) walking lifetime PA, (**C**) moderate lifetime PA, (**D**) vigorous lifetime PA, and (**E**) sedentary lifetime pattern across COVID-19 severity levels: comparing mean and standard error. An ANOVA test was conducted, revealing overall non-significant differences (*p* < 0.05).

**Table 1 healthcare-12-00813-t001:** Questionnaire validation scores given by health professionals.

Number of Experts	Relevance Mean ± SD	Clarity Mean ± SD	Writing and Terminology Mean ± SD	Format Mean ± SD	CVCt
11	4.53 ± 0.67	4.47 ± 0.77	4.44 ± 0.78	4.45 ± 0.82	0.90

Data are presented as mean and standard deviation. Validation scores range from 1 to 5. CVCt: total content validity coefficient.

**Table 2 healthcare-12-00813-t002:** Social and demographic characteristics of the study sample.

Variable	
Age (mean ± SD)	30.30 ± 12.05
Sex n (%)	
Female	130 (66.0)
Male	67 (34.0)
BMI kg/m^2^ (mean ± SD)	25.50 ± 4.21
Ethnicity n (%)	
Mexican Mestizo	197 (100)
Marital Status n (%)	
Single	123 (62.4)
Married	63 (32.0)
Consensual union	7 (3.6)
Other	4 (2.0)
Education n (%)	
Postsecondary education	137 (69.5)
Elementary and secondary education	60 (30.5)
Occupation n (%)	
Student	75 (38.1)
Employee	64 (32.5)
Self-employed	43 (21.8)
Unemployed	5 (2.5)
Other	10 (5.1)

Total sample = 197. Quantitative variables are represented as “mean (± SD)” and qualitative variables as “n (%)”.

**Table 3 healthcare-12-00813-t003:** COVID-19 general profile of the individuals.

Variable		Variable	
Diagnostic Tool n (%)		Days with Symptoms (mean ± SD)	8.57 ± 5.72
PCR	76 (38.6)	Hospitalized n (%)	
Antigen test	59 (29.9)	No	185 (98.4)
Antibody test	22 (11.2)	Yes	3 (1.6)
Antigen test + clinical	4 (2.0)	ICU	1 (33.3)
Antibody test + clinical	7 (3.6)	Oxygen Need n (%)	
Clinical	29 (14.7)	No	181 (96.3)
Year of Diagnosis n (%)		Yes	7 (3.7)
2022	56 (28.6)	Severity n (%)	
2021	81 (41.3)	Mild	91 (46.2)
2020	55 (28.1)	Moderate	89 (45.2)
2019	4 (2.0)	Severe	17 (8.6)
Symptoms n (%)		Post-COVID-19 Weight Loss (Kg) (mean ± SD)	1.48 ± 3.67
Present	188 (95.4)	Comorbidities n (%)	
Headache	149 (75.6)	No	162 (82.2)
Fatigue	137 (69.5)	Yes	35 (17.8)
Fever	119 (60.4)	Vaccination n (%)	
Sore throat	111 (56.3)	Yes	187 (96.4)
Anosmia	107 (54.3)	No	7 (3.6)
Cough	91 (46.2)	Initial vaccine type	
Joint pain	89 (45.2)	AstraZeneca	70 (37.4)
Eye pain	73 (37.1)	Sinovac	69 (37.0)
Nasal congestion	69 (35.0)	Pfizer	23 (12.3)
Runny nose	66 (33.5)	CanSino	15 (8.0)
Shortness of breath	60 (30.5)	Other	10 (5.3)
Diarrhea	26 (13.2)	COVID-19 Post Vaccine n (%)	
Vomit	11 (5.6)	Yes	62 (35.0)
Absent	9 (4.6)	No	115 (65.0)

Items totaling less than 197 (total sample) are attributed to instances where participants left items in blank. Quantitative variables are represented as “mean (± SD)” and qualitative variables as “n (%)”.

**Table 4 healthcare-12-00813-t004:** Significant Correlations Among Variables of Interest.

Variable	1	2	3	4
1. Age	1.00	0.45 **	0.22 **	0.17 *
2. BMI (Pre-COVID-19)		1.00	0.15 *	0.77 *
3. Days with Symptoms			1.00	0.17 *
4. Post-COVID-19 Weight Loss				1.00

* Significance at the 0.05 level (2-tailed), ** Significance at the 0.01 level (2-tailed).

## Data Availability

The data presented in this study are available on request from the corresponding author. The data are not publicly available due to privacy reasons.

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
