# Peer review of "The Role of Diet and Physical Activity in Shaping COVID-19 Severity: Design, Validation, and Application of a Retrospective Questionnaire"

_healthcare, 2024, doi:10.3390/healthcare12080813_

Round 1

Reviewer 1 Report

Comments and Suggestions for Authors

Dear Editor and authors, thanks for the opportunity to revise this manuscript. My only comment is about Table 2, which has an error in the title and first line. It seems to me it was layout. But I suggest checking it out.

Reviewer 2 Report

Comments and Suggestions for Authors

I recommend that authors try to amend their work according to the below suggestions.

Abstract:

Line 30. What do you mean by ‘’PA’’ in which is absent in walking PA? Physical activity? Please add the full name first and then abbreviation.

Methods:

How did you verify that the included participants had covid-19? Only through the questionnaire?

Did you do any power calculation?

Results:

Line 170: ‘’CVC’’, Please add the full name first and then abbreviation.

The participants who assessed the clarity and consistency of the questionnaire, were also part of the study sample? Did you consider their responses in the analysis?

Figure 2: To make it clearer, add to the x-axis ‘’diet’’ word to become ‘’excessive sugar diet’’ and Y-axis in figure 3

It would be a good idea if authors can look at the stratification of the data and consider the vaccine type and relate it to the study goal (perhaps in another study?)

Comments on the Quality of English Language

Minor editings

Reviewer 3 Report

Comments and Suggestions for Authors

The detail provided in this article, the statistical testing, and conclusions all are well articulated and presented for the reader. My concern about the conclusions reflect concern about the limitations of the survey used and the validity of conducting retrospective data collection on diet and physical activity. Retrospective questions about diet tend to reflect general patterns over time but not precise recall to a period that may have been years earlier. Although we know the typical period of having Covid, there is no indication of the average length of time (# of years or months) between taking the survey and the time period in question (pre-Covid). Other concerns about the survey questions:

·       The question about having a diet excessive in sugar is highly variable between people. One person’s excess is another’s limited use. There were no general qualifications included and to ask how many tablespoons were used (without giving a reference period of per meal, per day, etc.) seems irrelevant as a retrospective question.

·       The physical activity questions also concern me given the time lapse. The guidelines for using the IPAQ suggest that question wording or order not be changed. The questions used in this survey represent a change in both for most questions. The IPAQ asks about the week prior while this survey asks about PA during a week prior to getting Covid following an unknown time lapse that is dependent on when the subject had Covid. Recalling the average minutes spent doing exercise and the average intensity will be subject to recall bias. Additionally, the results reported focus on Lifetime PA and its intensity. Are these the questions related to the PA generally done pre-Covid diagnosis? 

·       Questions about walking equate walking around the house to someone doing power-walking. This seems to discount walking as an exercise.

The use of the survey for asking about diet and PA over a long time period is concerning. I recognize that all survey questions were validated but there is no clear discussion of the appropriate use of the questions in a retrospective manner. Even if respondents answered the questions similarly two times, there is not enough evidence to say that the recall over time is accurate. Did you analyze answers to the “lifetime” PA in comparison to the PA questions asked for the week prior to Covid? Were they similar (activity and intensity) and if so, did both show inverse associations with Covid severity?

In describing the analysis there is no mention about looking at different combinations of factors. For example, did you look to see if excessive sugar had a significant association alone only or if the association changed when combined with other dietary patterns (excessive red meat consumption). Also, does excessive sugar’s association change in combination with vigorous exercise?

Despite these concerns, I think that your conclusions do point to the need for additional research that could be conducted to validate the survey more accurately for retrospective studies and to do a more in-depth multi-factor analysis of the data. In your concluding paragraph you conclude that this is a validated survey, which is true to a limited degree, but could be expanded upon. This could require a study that has people record food intake more accurately for a week and then to have them recall that intake 6 months, 12 months, or 18 months later. Similarly, PA intensity can now easily be measured using wearable monitors that can accurately measure intensity and comparisons could be made to how respondents recall the intensity of their activity (short and long term recall).  Mentioning the need for further validation would strengthen this article.

Note: a word is missing in the beginning of line 221.
